# Assessing Stress Levels, Predictors and Management Strategies of Inmates at Ankaful Prison Complex in the Central Region, Ghana

**DOI:** 10.3390/bs13030201

**Published:** 2023-02-24

**Authors:** Edward W. Ansah, Jennifer Addae, John E. Hagan, Michael A. Baidoo

**Affiliations:** 1Department of Health, Physical Education and Recreation, University of Cape Coast, Cape Coast, Ghana; 2Neurocognition and Action-Biomechanics-Research Group, Faculty of Psychology and Sport Sciences, Bielefeld University, P.O. Box 10 01 31, 33501 Bielefeld, Germany; 3Department of Health Administration and Education, University of Education, Winneba, Ghana

**Keywords:** Ankaful Prison Complex, depression, inmates, management strategies, stress levels

## Abstract

Background: Stress among prison inmates is a neglected psychological health issue, but this phenomenon seems worse in Ghana’s prisons. This study examined the stress levels, predictors and management strategies utilized among inmates at Ankaful Prison Complex. Method: This survey sampled 1160 inmates using the census approach at the Ankaful Prison Complex with a self-developed questionnaire for the data collection. Frequency counts, one-way ANOVA, and multiple regression analysis were applied to the data. Results: The findings indicate that more than half of the inmates were moderately or highly stressed. Specifically, inmates at the Maximum Security Prison were the most stressed, followed by Annex Prison, Communicable Disease Prison, and the least, Main Camp Prison inmates. Inmate engagement in exercises, sporting activities, visit and chat with colleagues, and family connectedness outside the prison were stress-management strategies. Inmates’ self-reported stress levels were influenced by the prison of custody and state of depression. Conclusion: The moderate to high stress levels identified among inmates of Ankaful Prison Complex are influenced by person–environment factors. Management of the Ankaful Prison Complex is encouraged to initiate health screening services for inmates on common mental health challenges such as stress and to promote functional stress reduction interventions to improve prisoners’ mental health and overall well-being.

## 1. Introduction

Imprisonment, such as other forms of incarceration, is a major stressful experience in an individual’s life [1,2]. As a result of imprisonment and its associated challenges, inmates usually experience substantial modifications in their physical, psychological, and social functioning [3,4]. Despite inmates’ capacity to manage and adapt as humans, hurtful happenings may alter their biopsychosocial balance [2,5]. Imprisonment-related difficulties differ across a range of psychological disorders not limited to anxiety, depression, self-harm or other aggressive behaviours, obsessions, and psychoactive substance abuse [6,7,8]. Several authors [9,10,11] have argued that, beyond the physical and social conditions of prison establishments, psychological factors are important stressors that affect the health and well-being of inmates.

Epidemiological evidence thus far has documented a high prevalence of mental health issues or psychiatric morbidity among inmates in several countries, with a reported estimation higher than in the general population [12,13,14]. Available statistics show that nearly one in nine prison inmates suffer from common mental health challenges such as stress, depression and anxiety [13]. Thus, the prison environment is threatening and poses a high physical health and safety risk to even prison wardens. For some inmates, incarceration is emotionally painful to the extent that it creates a form of severe traumatic stress that causes post-traumatic stress reactions even after acquittal [15]. For instance, Haney [16] reported that at any particular point in time, a great amount of people with mental health problems are confined in the prison. Therefore, prisoners have higher physical and mental health needs above that of the general population [17]. Unfortunately, the occurrence of mental disorders, particularly stress, in prisons is great, but right of entry by health personnel to treat inmates is very little [18,19]. 

Stress as a psychological construct has become a global phenomenon to all people but is heightened among prison inmates. According to Zimmer, Wu and Domes [20], stress is seen as a specific association between individuals and their surroundings, considered by the individual as beyond available resources, and this inability threatens the person’s health and well-being. Due to stress sensation characterized by being burdened, wound-up, anxious, and apprehensive prison inmates usually face these characteristics while in custody because the conditions that exist are often alien to them (prisoners), and adapting to this environment becomes extremely burdensome [21]. O’Donovan, Doody and Lyons [8] found several types of stress such as acute, episodic acute, and chronic stress among inmates. These researchers further argued that acute stress has been the most identified form, the sources of which are the demands and pressures of the current, past as well as the anticipated demands and pressures of the future, such as spending time in incarceration. This type of stress is stirring and exciting in minor measures, but it becomes exhaustive when it is in excess or extended. However, with episodic types of stress, the victims take on too much and have too many challenges to deal with it. This situation typically occurs in long prison sentences. Individuals affected by stress are unable to consolidate the demands and pressures of stress and seem perpetually in the clutches of acute stress [22]. On the other hand, chronic stress is the crushing stress that drains people and negatively impacts the body, mind, and life of the individual.

Ankaful prisons is one of the local prisons and is the largest in Ghana. It is supposed to be responsible for the safe custody and welfare of inmates [23]. The prison complex has almost all the categories of inmates; however, its squalid conditions, poor food and overcrowding in the prisons [24] could have devastating effects on the physical health and mental well-being of these inmates. Despite the established link of stress and global burden of diseases, research generally on mental health of inmates across prison establishments, particularly in low-and middle- income countries such as Ghana, is sparse. Additionally, the prison situation in Ghana has been described as cruel, inhumane and degrading [25]. Thus, the prison setting may create an inhumane environment that is more likely to produce a high level of stress among inmates [26]. This study is drawn from the deprivation theory [26], which states that restrictions in the prisons could lead inmates to violence, aggression, anxiety, depression, stress, and suicide [26,27], hence the justification for this present study on the stress and associated antecedents among inmates. Furthermore, in Ghana, there seems to be no accurate account of inmates’ mental health-related challenges and associated health conditions. Therefore, understanding inmates’ inherent stressful experiences, health conditions and possible coping mechanisms are crucial towards designing appropriate mental healthcare programmes or interventions for this extremely vulnerable population. This phenomenon seems to be an unexplored research area to date in Ghana’s penal system. This study examined the predictors of stress and management strategies among inmates at Ankaful Prison Complex.

## 2. Materials and Methods

### 2.1. Research Design, Study Setting, and Participants’ Selection

This cross-sectional survey took place at the Ankaful Prison Complex, located in the Komenda-Edina-Eguafo-Abrem (KEEA) Municipal Assembly in the Central Region of Ghana. The prison complex houses persons with mental health challenges, short sentenced prisoners, convicts, prisoners on trial, remand prisoners, communicable disease prisoners and long sentenced prisoners, most aggressive prisoners, and hardened as well as high-profile prisoners [23]. The complex has a prisoner population of 1491, of whom 84% are literate while 16% have no formal education. The complex houses: Maximum Security Prison (N = 649), Main Camp Prison (N = 236 prisoners), Ankaful Prison Annex (N = 541 prisoners) and a Communicable Disease Prison (N = 64 prisoners), as of 30 March 2019 [23]. 

This study targeted all (1491) the prisoners in the complex using a census sampling approach; however, 1160 (approximately 78%) inmates completed the instrument. Specifically, we collected 497 data points from the Maximum-Security Prison, 382 from Ankaful Prison Annex, 221 from the Main Camp Prison, and 60 from the Communicable Disease Prison. The outcome of the sample was extrapolated to reflect the stress levels of all prisoners in the Ankaful Prison Complex. The unit of analysis was the individual prisoner or inmate. By using this sampling approach, potential sampling errors were dealt with since there was no need to select a sample from the prisoners. 

### 2.2. Instrumentation

A questionnaire, developed from pre-existing instruments, was used for data collection. We measured stress from the Perceived Stress Scale [28], health status from the Short Form Health Survey [29] and depression from Hamilton’s Depression Rating Scale [30]. The questionnaire also solicited demographic variables of the inmates such as age, education, religion, number of years in prison, number of years sentenced, number of children, prison history and number of visitors they receive. 

A 10-item perceived stress scale [28] was used to measure the stress levels of the inmates. In scoring, items, 4, 5, 7, and 8 were reversed; 0 = 4, 1 = 3, 2 = 2, 3 = 1 and 4 = 0. The stress levels of the inmates are categorized as low (1–17), moderate (18–33) and high (34–50), using the aggregated scores. Respondents rated the frequency of their feelings related to events and situations that occurred in the prison in the last month. The ratings were on a five-point Likert-type scale (1 = never to 5 = very often). A higher score means greater stress, and to produce such scores, the four positive items were reverse-scored, after which all the items were summed up to produce scores from 0 to 40. Moreover, the scale recorded a reliability value of 0.83 [31]. Some of the items are “*in the last month, how often have you been upset because of something that happened unexpectedly?” and “in the last month, how often have you felt that you were unable to control the important things in your life?*”.

The 9-item Short Form Health Survey measures the health status of participants on a six-point scale (1 = very poor to 6 = excellent). A lower score means the presence and compromised health status of the inmates [29]. A sample item was “*overall, how would you rate your health during the past 4 weeks?*” These items recorded reliable composite (0.92) and alpha (0.86) reliability [32]. Moreover, Hamilton’s Depression Rating Scale [30], which has 17 items with scores ranging from 0 to 17 as a normal range, whereas 20 or more indicates a moderate to severe depression state. These items related to characteristics such as depressed mood and feelings of guilt. Participants responded by scoring items 0–4, where 0 means not depressed, and 4 means highly depressed. A recent review indicated that the scale is reliable [33]. 

### 2.3. Ethical Statement and Study Procedure

Ethical clearance was approved by the Institutional Review Board of the University of Cape Coast (ID: UCCIRB/CES/2019/03) for this study. Additionally, authorization or approval was sought from the headquarters of the Ghana Prison Service and the Commanders of the Ankaful Prison Complex for the data collection. The questionnaire was administered to the prison inmates at the Maximum-Security Prison, Main Camp, Annex Prison and Communicable Disease Prisons. Furthermore, we assured the inmates of their anonymity, confidentiality and their voluntary participation, and that they could withdraw from the study at any point in time without a consequence. Moreover, each participant gave a verbal or written consent before taking part in the study. Prison peer educators were trained on the interpretation of the questionnaire. The peer educators who were also fluent in the predominantly spoken local dialect (e.g., Twi) assisted other inmates who could not read and write the English language. Data collection was carried out by the researchers aided by 30 peer educators of the various prison establishments. The questionnaire was administered to the inmates at the four prison establishments after mid-day ration, between 12 and 2 pm. Data collection took about two weeks. We paid no monetary incentives to any participant.

### 2.4. Statistical Analyses

Responses to the questionnaire were edited, coded, scored and entered into SPSS version 20.0 software. Preliminary normality check using Levene’s test of homogeneity of variance was calculated (Levene *St*. = 42.58, *p* = 0.001) which revealed that the date was normal. Thus, frequency and percentage counts were applied to the data in determining the stress levels of the inmates. In addition, one-way analysis of variance (ANOVA) was calculated to differentiate the levels of stress according to the four prison establishments holding these inmates. Furthermore, we applied Bonferroni post hoc test for multiple comparison of stress levels among the various prison establishments. In addition, multiple regression was used to test the extent to which depression, unit of prison holding the inmates, age, education, marital status, nationality, religion and migraine influence the stress levels of the inmates. The scores for health status, level of stress and depression were aggregated scores while stress-coping strategies used the individual items because they are nominal. Initial check indicated that the predictor variables were not highly correlated; thus, the enter method was applied in the analysis. 

## 3. Results

### 3.1. Socio-Demographics and Stress Level of the Inmates

The 1160 prison inmates in this study comprised 42.8% (497) prisoners from the Maximum Security Prison, 32.9% (382) from the Annex, 19.1% (221) from Main Camp and 5.2% (60) from the Communicable Diseases Prison. About 33.6% (390) of the inmates were between the ages of 18 and 30, 33.5% (389) 31 and 40 years, 21.3% (247) 41 and 50 years, 7.7% (89) 51 and 60 years, while 3.9% (45) was above 60 years. Furthermore, 44.0% (510) had basic education or less, 27.2% (315) secondary education, 20.9% (242) non-formal education, and 8.0% (93) tertiary education. Moreover, 68.8% (798) were Christians, 25.9% (300) Muslims, whereas 5.3% (62) were Traditionalists. Close to one third, 66.8% (774) of the inmates were serving a prison sentence between 1 and 20 years, 13.4% (115) 20 and 50 years, 9.0% (104) less than 1 year, 5.2% (60) 51 and 100 years, 3.1% (36) were awaiting trial, 1.5% (17) 101 and 150 years, 0.8% (9) lifetime imprisonment, and 0.4% (5) 151 years and above. The rest of the characteristics are presented in Table 1 below.

Frequency results indicate that out of 1160 inmates, 3.4% (n = 39) recorded a low level of stress, 90.3% (n = 1048) moderately stressed, while 6.3% (n = 73) a high level of stressed. Hence, the majority of the inmates at Ankaful Prison Complex are either moderately or highly stressed, a situation that is expected. Many of the inmates also use physical exercise (44.1%), religious activities (24.5%), and interacting with other inmates (11.2%) as coping strategies. Unfortunately, a few other also engaged in unhealthy coping behaviours such as oversleeping (4.3%), use of hard drugs (2.7%), self-isolation (2.1%), bulimia (1.6%), and hunger strike (0.8%).

### 3.2. Differences in the Stress Levels of Prison Inmates 

ANOVA results revealed a statistically significant difference among the inmates from the various units, *F* (3, 1156) = 29.86, *p* = 0.001. In testing the effect size, Bonferroni post hoc analysis revealed that inmates at the Maximum-Security Prison (M = 27.41, SD = 4.57) were highly stressed compared with those at the Main Camp (M = 26.45, SD = 5.23), while inmates at the Annex reported higher levels of stress than their counterparts at the Main Camp (M = 26.45, SD = 5.23). Again, inmates at the Communicable Disease Unit were highly stressed compared with those at Main Camp (M = 26.45, SD = 5.23) (see Table 2). Therefore, the stress levels of inmates at the Ankaful Prison Complex are different and are dependent on the unit of prison that holds them. 

### 3.3. Predictors of Stress Level of Prison Inmates 

Multiple regression model revealed statistical significance in determining the stress levels of the inmates at the Ankaful Prison Complex, *F* (8, 706) = 50.34, *t* = 23.24, *p* = 0.001, accounting for about 20% of change in the stress levels (see Table 3). However, only depression, *t* (8, 706), = 17.76, *p* = 0.001, and units of prison holding the inmates, *t* (8, 706), 2.16, *p* = 0.001, were significant in predicting their stress levels. Therefore, the stress levels of inmates at Ankaful Prison Complex are influenced by their level of depression and the unit of prison establishment holding them. 

## 4. Discussion

This study assessed stress levels, predictors and management strategies in a sample of prison inmates in the Central Region of Ghana. Generally, over 96% of the inmates at Ankaful Prison Complex were either moderately or highly stressed. This finding could be attributed to the historical reasons (and continuous implementation of such) for the establishment of prisons and correctional centres, where prison facilities were initially created to punish individuals for their wrongdoings in society [10]. In addition, prison sentences place limitations and restrictions on inmates, coupled with the “criminal” tag society associates with every prisoner. Unfortunately, prison set-ups in Ghana are purposely built to make inmates stressed to deter them from criminal acts and to save society. Therefore, the prison atmosphere is likely to be highly stressful to the inmates. For instance, Pollock [34] contends that atmospheres within several prison facilities are toxic, and this situation affects prisoners’ material possessions and freedom. Accordingly, over half of inmates can earn little or no income while incarcerated, many lose their job or livelihood, spend their life savings, and have their total lifetime earning capacity affected. Moreover, the stress levels become heightened when inmates have to always be under control of the security guards. These experiences could be classified as the worst form of punishment ever created by society, making them extremely stressful. 

The poor conditions of Ghana’s prison establishments, including the long sentences and harassment, can also create high levels of stress to inmates. According to Hall et al. [35], stress varies with the analysis of a situation that a person is experiencing, suggesting that stressful situations depend upon an individual’s experience in life. The results suggest that individual’s interactions with the various environmental conditions (P-E fit) surrounding them affect their life and well-being [36]. Moreover, observations indicated that environmental conditions such as prison threaten individuals’ status, beliefs, and self-esteem, among others [37]. Therefore, inmates who are exposed to poor living, which hitherto was not so, are likely to be under increased stress. Furthermore, Machator [38] observed that Ghanaian prisoners go through serious stress during incarceration due to unfavourable conditions, similar to Nigerian prisons [39], where poor conditions within the establishments have strong associations for personal harm and adaptation difficulties for the inmates [40,41].

Although these inmates reported high levels of stress, it is also dependent on the prison establishment holding them. This may be because the holding unit is assigned according to the crime committed and period of sentence given. For instance, the Maximum-Security Prison incarcerates high-profile cases such as murder and armed robbery perpetrators, resulting in a long period of imprisonment, a situation which could make such inmates more stressful than others. It was also observed that some inmates are serving prison terms between 50 and 150 years, and the longer they were there, the more they experienced the prison’s negative conditions with the associated stress. The results further suggest that inmates in the Annex Prisons are more stressed than their colleagues in the Main Camp and Communicable Disease Prisons. Inmates in Annex Prisons are either on remand or on trial, that is, they are yet to be convicted. Therefore, they are not certain about their future and are always in suspense regarding their status. In addition, such inmates are always moving between the prison and the courts, where they interact with lawyers, judges and others connected to their cases. The continuous holding of such inmates in prison custody places psychological burden on them (i.e., heightened stress level), which compromises their health condition. Evidence indicated that the remand prison environment poses high stress because such inmates are uncertain about their future [42]. This outcome means that such inmates would need improved psychological therapy to protect their health. 

The stress level among inmates in the Maximum-Security Prison, Annex Prisons, and Communicable Disease Prison were higher than that of inmates at the Main Camp Prison. This may be because inmates at the Main Camp have more interactions with outside society. Furthermore, Main Camp Prisoners have a fewer number of periods (about six months) left to serve in their prison terms to be free from imprisonment. In other words, such inmates have a shorter time until the end of their prison terms, and that may reduce their stress levels because of the anticipated freedom. Moreover, the provision of extension services (offering of labour outside the prison), by Main Camp Prison inmates, to some extent, de-stresses and makes them feel “whole” again. The feeling that they are about to leave the confines of the prison could also provide a source of joy and relief [43]. These inmates are seen as jovial. The net effect is that their level of stress is significantly lower compared with those who have more years to serve, and others who do not know their fate. This confirms Wayne’s [43] thoughts that inmates who have regular interactions with society may have very little stress or burdens, reduced perceived social segregation, and a lesser feeling to compete to fit into society. 

It was further observed that depression but not anxiety had a significant influence on the level of stress of inmates. A sentence to prison is among the most stressful and depressing events in a prisoner’s life. Imprisonment is a traumatic experience for a person, as it restricts one’s liberty of movement. Thus, inmates are under a high level of stress mentally and physically, leading to psychological changes that lead to depression [44]. Incarceration can largely affect the thinking and behaviour of an inmate and cause austere depression [45]. However, the psychological effect of depression on each inmate differs with respect to duration of sentence, condition, and place held [44]. Thomas also indicated that depression is more likely to occur in an inmate subjected to pain, fear, and loneliness. Therefore, the ability of the inmates to adopt effective management and coping strategies in reducing their stress and depression levels could be valuable. Thus, inmates need healthy coping strategies to reduce their level of depression and stress. 

A number of coping strategies including bulimia, self-isolation, use of hard drugs, oversleeping and refusal to eat, reported by the inmates, which are unhealthy and maladaptive behaviours that are likely to have debilitating effects on their health and well-being. Perhaps, the inmates may not know if the strategy for coping with stress is creating more harm than good to themselves. This means that inmates may be dealing with stress in a way that could have long-term negative effects on their health. Fortunately, many inmates engaged in activities like exercises/sports, religious activities, chatting with colleagues, singing, calls home or receive visitors, which could have a positive influence on their health and well-being [46,47]. Activities like exercises/sports help improve psychological states such as mood, anxiety and physical health of inmates or isolated individuals. Moreover, visitation from family and friends as well as chatting with colleagues eliminate the feelings of isolation or abandonment and provide some assurance of belongingness to the inmates. Thus, such activities need to be encouraged.

Further findings revealed that stress levels of the inmates did not influence their health status. We believe that this is a methodological issue because depression and stress were entered into the regression model together. Depression is a more severe form of stress [48]; hence, depression can suppress the effect of stress levels of inmates on their health status. Thus, it may not be methodologically appropriate to measure and test the depression and stress level of participants in combination, in which case the inmates would have graduated from stress to depression and that testing the two constructs in one model rendered the lower one (stress level) as nonsignificant. This is supported by Shrestha et al. [45] and Ahmad and Mazlan [6] in that episodes of depression are typically identified among inmates after prolonged stressful experiences.

### Limitations and Future Directions 

This study has some limitations; hence, the findings identified should be interpreted with caution. The cross-sectional design used limits to define the causal linkage between stress levels, health status and the identified factors. Further, data gathered were self-reported based on past histories or experiences by the inmates, which are subject to recall biases. For example, inmates are more likely to exaggerate their stressful experiences; hence, they may have overestimated the phenomenon. Future research should consider a multi-site study using novelty approaches to data collection such as the ecological momentary assessment to provide more accurate information on the level of stress and other mental disorders such as depression and associated factors not captured in the present study across different prisoner populations in Ghana. Such studies could facilitate comparative analyses for appropriate multi-site-specific interventions. 

## 5. Conclusions

The inmates at Ankaful Prison Complex are exposed to high levels of stress due to the various stressors. However, the inmates have ways of handling their stress. In addition to the activities organized by the management of Ankaful Prison Complex, many of the inmates engage in physical activities, visit friends and chat with them, sing, and engage in some religious activities as ways to reduce their stress levels. Unfortunately, others become isolated, use hard drugs and overeat to limit stress to address their stressful situations. Moreover, inmates’ stress levels depend on the prison of custody, and the longer the sentence term, the higher the stress level. Generally, although stress usually leads to depression, stress levels of the inmates did not influence their health status, but depression markedly affected their stress levels and health status. The needed psychological health interventions should be structured to minimize or eliminate the stress levels of the inmates according to the prison establishment holding them. Such interventions should be situation-specific to prison establishment.

## Figures and Tables

**Table 1 behavsci-13-00201-t001:** Demographic characteristics of inmates of Ankaful Prison Complex.

Variable	Frequency	Percentage (%)
Name of Prison		
Maximum Security	497	42.8
Ankaful Prisons Annex	382	32.9
Main Camp	221	19.1
Communicable Diseases Prison (CDP)	60	5.2
Age (yrs)		
18–30	390	33.6
31–40	389	33.5
41–50	247	21.3
51–60	89	7.7
Above 60	45	3.9
Education		
Basic	496	42.8
Secondary	315	27.2
Non-formal	242	20.9
Tertiary	93	8.0
No formal education	14	1.2
Religion		
Christians	798	68.8
Muslims	300	25.9
Traditionalists	62	5.3
Years of sentence (years)		
Awaiting trial	36	3.1
Less than 1 year	104	9
1–20	774	66.8
20–50	115	13.4
51–100	60	5.2
101–150	17	1.5
151 years and above	5	0.4
Lifetime	9	0.8
Years spent in prison (years)		
Less than a year	268	23.1
1–5	440	37.9
6–10	232	20
11–15	123	10.6
16–20	66	5.7
21–25	15	1.3
26–30	8	0.7
30 years and above	8	0.7
Prison History of inmate		
First time	948	81.7
Imprisoned before	212	18.3
Visitors		
Received visitors	723	62.3
No visitors received	437	37.7
Coping Strategies		
Exercise and Sporting activities	511	44.1
Religious activities	285	24.5
Vising and chatting with colleagues	130	11.2
Calling the family	74	6.4
Sleeping	50	4.3
Using hard drugs	31	2.7
Isolating oneself from others	27	2.3
Singing	24	2.1
Eating a lot	19	1.6
Refusal to eat	9	0.8

**Table 2 behavsci-13-00201-t002:** Differences in inmate stress levels by their units of prison.

Prison Unit	N	M	SD	*df*	*F*	Sig.
				3, 1156	29.86	0.001
MSP	497	27.41	4.57			
Ankaful PA	382	26.88	2.91			
Main Cam P	221	26.45	5.23			
Comm. DP	60	26.57	5.49			

Key: MSP—Maximum Security Prison, Ankaful PA—Ankaful Prisons Annex, Main Cam P—Main Camp Prison, Comm. DP—Communicable Diseases Prison.

**Table 3 behavsci-13-00201-t003:** Effects of depression, migraine, prison unit, demographics of inmates on their stress levels.

Variable	B	Βeta	*R*	*R* ^2^	*t*	Sig.	Tol.	VIF
Constant	19.87		0.61	0.37	23.24	0.001		
Depression	0.589	0.584			17.76	0.001	0.962	1.04
Prison Unit	−0.929	−0.173			−5.29	0.001	0.847	1.18
Age	0.263	0.061			1.73	0.084	0.740	1.35
Education	0.085	0.022			0.73	0.466	0.993	1.01
Marital Status	0.016	0.003			0.09	0.922	0.827	1.21
Nationality	0.238	0.034			1.07	0.286	0.916	1.01
Religion	−0.368	−0.048			−1.54	0.124	0.929	1.08
Migraine	0.030	−0.077			−0.56	0.577	0.934	1.07

*df* (8, 706), *F* (50.34).

## Data Availability

The data are deposited and available here: osf.io/b49ta (accessed on 19 January 2023).

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
