# Peer review of "Assessing Stress Levels, Predictors and Management Strategies of Inmates at Ankaful Prison Complex in the Central Region, Ghana"

_behavsci, 2023, doi:10.3390/bs13030201_

Round 1
Reviewer 1 Report
Dear Authors,
Thank you for the opportunity to revise this manuscript.
First of all, I want to congratulate the authors for carrying out such a robust study.
This manuscript is convincingly ambitious, sound, and credible, and has soundness methodology.
- The topic of this manuscript is interesting, and it introduces an innovative aspect. The theme is relevant and has a conceptual domain, the study is a well-conceived, well-crafted, and well-presented paper.
- The methods are appropriate, accurate, and objective for the experiment, and improve the understanding of the reader.
- The analyses are not complex, informative, and appropriate for the experiments.
- The authors recognize the limitations of this research, and in my view, this manuscript has main value and provides evidence that the authors seem to acknowledge the main innovation of their study,
- Overall, the paper is well-placed to stimulate future research within the field.
- It is clear a substantial amount of work has gone into preparing this manuscript.
- Please provide the gender of participants, the range of age (minimum and maximum ), and standard deviation (SD) in the participant's section,
- The discussions and conclusions do not address all the results of the study. Please improve it
- Please provide after the discussions the limitations and future directions in a section denominated “Limitations and future directions”
- The citations in the manuscript are not correct e.g. line 39 [6,7,8] must be [6-8]; [9,10,11] must be [9-11]. Please revise the entire article
There are inconsistencies in the references. Please correct the full citations and references according to the journal guidelines.
I hope my comments will be helpful in the process of revising this manuscript.
Author Response
We are grateful for the review and the opportunity for rework and resubmit our manuscript.
Please see the attachment.

Reviewer 2 Report
This is an interesting and useful manuscript. Some comments...
INTRODUCTION: This section is succinct and clear. However, acknowledging major theories of the prison environment itself as a stressor (e.g., deprivation theory; Sykes, 1958) would have better located this piece. Otherwise, the rationale is appropriate and well-phrased.
METHODS: This section is also clear and the authors disclose the decisions as to their approach. The sample is well-defined and it is a credit to the researchers that you were able to amass such a solid return on your survey/questionnaires. This is no small feat in a prison context. Re: measures. On the face of it, the selected measures are appropriate for the variables of interest. However, it would be good to make explicit whether these tools have normative data to support their use in both prisons as well as Ghana.
RESULTS: This section is quite straightforward. The data is well-presented and clear to the reader. Conclusions appear to be supported by the data. Although this is not an extensive section, the inclusion of headings would have made this section even easier to navigate, especially because so much data is presented in the body of the text as well.
DISCUSSION/CONCLUSIONS: Again, this section is well presented and argued. Main findings are adequately synthesised to address the research question. Limitations are noted and the point about prisoner reliability (or lack thereof) is fair in the context of research (or any official information-gathering context).
GENERAL: This is a useful addition to the literature, especially the African correctional literature. Although the writing tone is appropriate on the whole, there are some occasional style differences (e.g., "may be feeling a bit happy" line 263). Also, the authors should be careful about 'othering' language (i.e., 'they/them' - see line 290). Efforts should be made to use inclusive language.
Author Response
We thank the reviewer for the review done.
Please see the attachment.
